# AnyMS: Bottom-up Attention Decoupling for Layout-guided and Training-free Multi-subject Customization

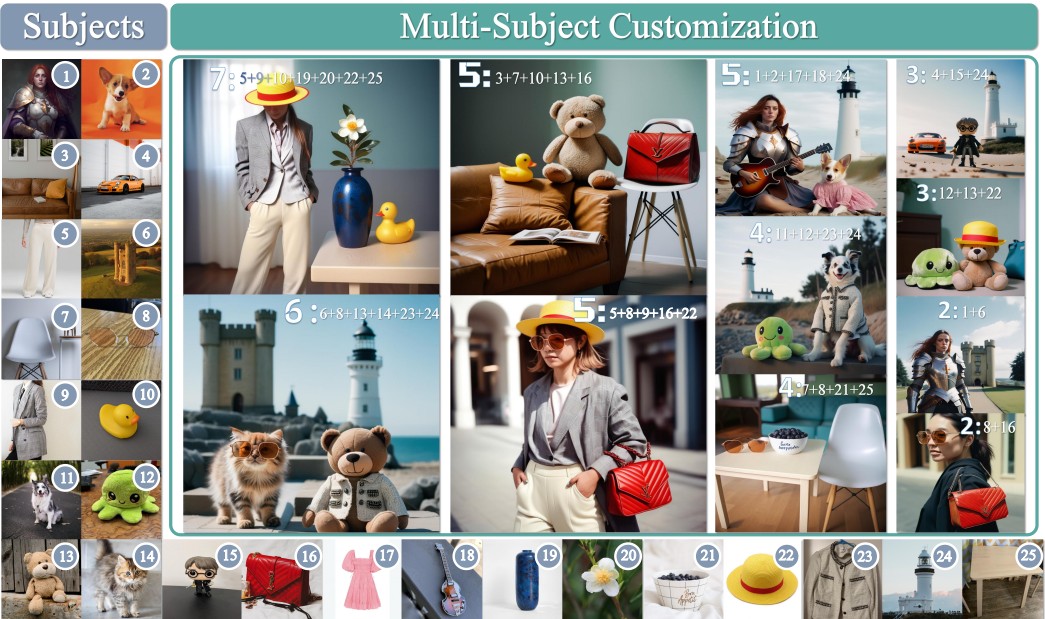

Figure 1: AnyMS enables **training-free** layout-guided multi-subject customization, supporting diverse subject combinations and scaling to larger numbers while maintaining a balance among layout control, text alignment, and identity preservation. See more visualization details and layout configurations in the Appendix.

## Abstract

Multi-subject customization aims to synthesize multiple user-specified subjects into a coherent image. To address issues such as subjects missing or conflicts, recent works incorporate layout guidance to provide explicit spatial constraints. However, existing methods still struggle to balance three critical objectives: text alignment, subject identity preservation, and layout control, while the reliance on additional training further limits their scalability and efficiency. In this paper, we present **AnyMS**, a novel training-free framework for layout-guided multi-subject customization. AnyMS leverages three input conditions: text prompt, subject images, and layout constraints, and introduces a bottom-up dual-level attention decoupling mechanism to harmonize their integration during generation. Specifically, global decoupling separates cross-attention between textual and visual conditions to ensure text alignment. Local decoupling confines each subject's attention to its designated area, which prevents subject conflicts and thus guarantees identity preservation and layout control. Moreover, AnyMS employs pre-trained image adapters to extract subject-specific features aligned with the diffusion model, removing the need for subject learning or adapter tuning. Extensive experiments demonstrate that AnyMS achieves state-of-the-art performance, supporting complex compositions and scaling to a larger number of subjects. Our project is available at this anonymous link https://anonymous.4open.science/r/AnyMS-659CAJDFQMYBH

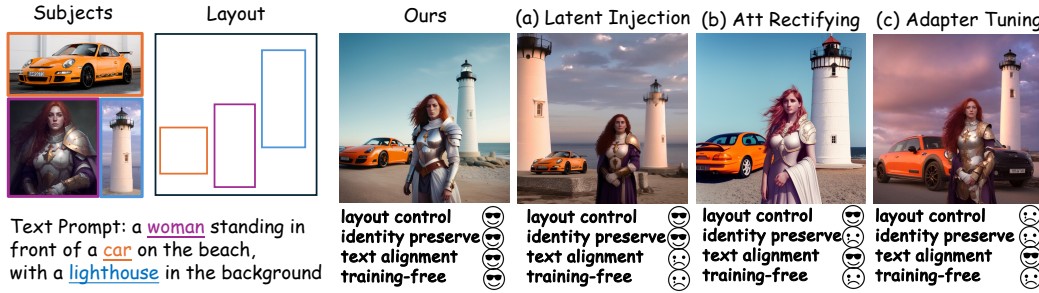

Figure 2: **Layout-guided Multi-subject Customization Results.** (a) Result of latent injection method MuDI (Jang et al., 2024). (b) Result of attention rectifying method Cones2 (Liu et al., 2023b). (c) Result of adapter tuning method MS-Diffusion (Wang et al., 2024a). Different colors show the associations between subjects and their corresponding layout configurations.

# 1 INTRODUCTION

Recent advances in large-scale pre-trained diffusion models (Dhariwal & Nichol, 2021; Nichol et al., 2021; Ramesh et al., 2022; Rombach et al., 2022; Saharia et al., 2022) have enabled the novel application of customized image generation (Gal et al., 2022; Ruiz et al., 2023; Chen et al., 2023; Wang et al., 2024b), allowing users to generate images containing specific subjects of interest. While single-subject customization has achieved remarkable success (Gal et al., 2022; Ruiz et al., 2023; Gal et al., 2023; Li et al., 2024), recent research has advanced toward the more challenging task of multi-subject customization (Kumari et al., 2023; Liu et al., 2023a; Gu et al., 2024). This paradigm focuses on synthesizing multiple custom subjects into coherent scenes guided by textual prompts, thereby offering greater flexibility and personalization in user-driven content creation.

Mainstream multi-subject customization methods have evolved through several approaches. A common line of work adopts joint training (Kumari et al., 2023; Liu et al., 2023a) with data augmentation (Han et al., 2023; Jang et al., 2024), optimizing the model on mixed multi-subject data. Alternatively, some methods leverage single-subject tuning with composition strategies (Kong et al., 2024; Kwon et al., 2024; Jin et al., 2025), where individually learned subjects (*e.g.*, LoRAs (Chen et al., 2025)) are combined to form multi-subject scenes. While effective in simple settings (*e.g.*, two subjects), these approaches struggle to scale as the number of subjects increases, often resulting in subject missing or conflict, *i.e.*, some subjects fail to appear, or their attributes become confused. Subsequently, a new direction (Liu et al., 2023b; Gu et al., 2024; Wang et al., 2024a; Zhu et al., 2025) has emerged to introduce additional layout guidance (*e.g.*, bounding boxes), which aims to alleviate the above issues by explicitly constraining the spatial arrangement of subjects.

Specifically, existing layout-guided multi-subject customization methods mainly implement the layout guidance from three perspectives: **a) Latent Injection.** The typical method (Jang et al., 2024) composes segmented subjects within bounding boxes to form an initial latent noise, injecting spatial and appearance priors. While effective for layout and identity preservation, such latent-level constraint often undermines text alignment (*e.g.*, object relations), resulting in incoherent generation (*c.f.*, Figure 2(a)). **b) Attention Rectifying.** These methods (Liu et al., 2023b; Chen et al., 2024; Zhu et al., 2025) adjust the cross-attention map of each subject using bounding boxes, typically by enhancing activations in target regions while suppressing irrelevant areas. However, this attention-level guidance often requires encoding each subject as a special token concatenated with the text prompt and controlled via text cross-attention, which can cause conflicts between visual and textual conditions, leading to imprecise identity preservation shown in Figure 2(b). **c) Adapter Tuning.** MS-Diffusion (Wang et al., 2024a) introduces a trainable adapter module to jointly encode visual features, text embeddings, and layout constraints. However, they rely on carefully curated layout-labeled multi-subject data with additional module tuning, which significantly increases computational cost and limits their generalization capability to unseen subjects or combinations (*c.f.*, Figure 2(c)). In summary, existing layout-guided approaches still suffer from two major limitations:

- **1)** Difficulty in balancing the trade-off among text alignment, subject identity preservation, and layout control, especially as the number of subjects increases.
- **2)** Reliance on additional training for subject learning or adapter tuning, leading to strong data dependency and substantial computational overhead.

To address these limitations, we propose **AnyMS**, a novel *training-free* layout-guided multi-subject customization framework. Based on the three types of input conditions for layout-guided multi-subject customization — *textual* (*e.g.*, text prompts), *visual* (*e.g.*, subject images), and *layout* (*e.g.*, bounding boxes) — AnyMS performs a bottom-up dual-level attention decoupling to balance their integration alongside the general denoising process of diffusion generation: **1)** *Global decoupling*: separating cross-attention between text (*i.e.*, text cross-attention) and subject images (*i.e.*, image cross-attention) to mitigate global conflicts between textual and visual conditions, thereby ensuring text alignment. **2)** *Local decoupling*: further disentangling image cross-attention using layout constraints, where each region only attends to its corresponding subject, avoiding interference among multiple subjects, thus guaranteeing both subject identity preservation and layout control.

In addition, AnyMS employs pre-trained image adapters (Ye et al., 2023; Li et al., 2024) to extract subject-specific visual features aligned with the diffusion model, thereby eliminating the need for time-consuming subject learning or additional tuning. By decomposing and harmonizing textual, visual, and layout conditions, AnyMS achieves a better balance between text alignment, subject identity preservation, and layout control (*c.f.*, Figure 2). Extensive experiments show that our method achieves state-of-the-art performance, supporting more complex and creative multi-subject customization with improved efficiency and generalizability.

In summary, we made three contributions in this paper: 1) We introduce **AnyMS**, a novel *training-free* framework for layout-guided multi-subject customization that employs a bottom-up dual-level attention decoupling mechanism that disentangles textual, visual, and layout conditions. 2) We incorporate pre-trained image adapters to efficiently extract subject-specific features, eliminating the need for additional tuning or subject learning. 3) We conduct extensive experiments across diverse benchmarks, demonstrating that AnyMS achieves state-of-the-art performance with improved efficiency, supporting complex compositions and scaling to a larger number of subjects.

## 2 RELATED WORK

**Text-to-image Generation.** Text-to-image (T2I) generation aims to synthesize realistic images from textual descriptions. Early GAN-based approaches (Reed et al., 2016; Li et al., 2019) have been largely surpassed by diffusion models (Ho et al., 2020; Song et al., 2020), which progressively denoise latent variables under text guidance (Dhariwal & Nichol, 2021; Ho & Salimans, 2022). Recent advances such as Stable Diffusion (Rombach et al., 2022) and SDXL (Podell et al., 2023) further improve fidelity and efficiency. While these models achieve remarkable performance on generic prompts, they cannot directly handle more complex cases of user-specific customization.

**Single-subject Customization.** Customized generation introduces user-specified subjects into T2I models with faithful identity preservation and textual description alignment. Existing single-subject customization approaches can be categorized into three groups: (1) *Training-based*, such as Dream-Booth (Ruiz et al., 2023) and Textual Inversion (Gal et al., 2022), which fine-tune or optimize embeddings to bind subjects with special tokens or parameters; (2) *LoRA-based*, where lightweight parameter tuning (*e.g.*, LoRA (Soboleva et al., 2025; Kong et al., 2024)) is employed to inject subject-specific features; (3) *Adapter-based*, such as Blip-Diffusion (Li et al., 2023) and SSR-Encoder (Zhang et al., 2024), which learn auxiliary encoders to extract subject features aligned with diffusion models. These approaches are effective for a single subject but face scalability challenges when extended to multiple subjects.

**Multi-subject Customization.** The multi-subject setting requires generating coherent images with multiple customized subjects, where the main difficulty lies in disentangling subject features and maintaining prompt alignment. Training-based approaches, such as CustomDiffusion (Kumari et al., 2023) and MUDI (Jang et al., 2024), employ joint optimization or data augmentation to improve disentanglement, but suffer from high cost and limited generalization. Inference-time methods (Jiang et al., 2025; Jin et al., 2025; Kwon & Ye, 2024; Ding et al., 2024), like TweedieMix (Kwon & Ye, 2024), and FreeCustom (Ding et al., 2024), instead manipulate latent variables or attention maps to merge learned subjects without retraining. While reducing overhead, these methods often encounter subject omission, attribute confusion, or degraded fidelity as the number of subjects grows.

**Multi-subject Customization with Layout Control.** To further mitigate conflicts among multiple subjects, recent studies introduce layout guidance (*e.g.*, bounding boxes) to constrain spa-

tial arrangement. Existing methods can be grouped into three paradigms: (1) *Latent injection*: MuDI (Jang et al., 2024) initializes latent codes from segmented subjects within bounding boxes, while OMG (Kong et al., 2024) leverages layouts from non-customized images for spatial priors; (2) *Attention rectifying*: Cones2 (Liu et al., 2023b) and Mix-of-Show (Gu et al., 2023) enforce attention activations within target regions, helping disentangle object features; (3) *Adapter tuning*: MS-Diffusion (Wang et al., 2024a) introduces an adapter to jointly encode subject, prompt, and layout inputs, but requires layout-labeled training data. Although effective in improving spatial controllability, these methods either incur heavy training costs or struggle to balance text alignment, identity preservation, and layout control. In contrast, our AnyMS achieves a better balance across the three objectives in a training-free manner, and scales naturally to complex compositions.

## 3 METHOD

### 3.1 PRELIMINARY

**Stable Diffusion Model.** Stable Diffusion Model is the text-to-image representative generative model consisting of three main components: an autoencoder $(\mathcal{E}(\cdot), \mathcal{D}(\cdot))$, a denoising network $\varepsilon_\theta(\cdot)$ and a text encoder CLIP $\tau_\theta(\cdot)$ (Radford et al., 2021). Given an image $x$ and a text prompt $P$, the autoencoder maps the image from the pixel space to the latent space $z_0 = \mathcal{E}(x)$, while the CLIP encoder encodes the prompt to the conditional embedding $c_t = \tau_\theta(P)$. For the forward process, starting from $z_0$, sampled random Gaussian noise $\varepsilon \sim \mathcal{N}(0,1)$ is applied to $z_0$ to get $z_t = \sqrt{\bar{\alpha}_t} z_0 + \sqrt{1 - \bar{\alpha}_t} \epsilon$ at timestep $t$, where $\alpha$ is predefined coefficient provided by noise scheduler. For the backward process, the diffusion model is trained conditioned on the current latent $z_t$, timestep $t$ and text prompt conditions $c_t$ to predict the added noise. The model is trained with the following reconstruction loss:

$$L_{\text{rec}} = \mathbb{E}_{z,\varepsilon \sim \mathcal{N}(0,1),t,c_t} \| \varepsilon - \varepsilon_\theta (z_t, t, c_t) \|_2^2 \tag{1}$$

$z_t$ is progressively denoised to obtain $z_0$, after which the decoder maps it back to pixel space $x = \mathcal{D}(z_0)$.

**Cross Attention Mechanism** Stable Diffusion Model utilizes U-Net (Ronneberger et al., 2015) as the backbone, which contains the cross-attention modules to integrate the text prompt. Specifically, given the latent features $Z$, the output $Z_{\text{out}}$ of cross-attention mechanism can be formulated as:

$$Z_{\text{out}} = \text{CA}_{\text{text}}(Z, P) = \text{Softmax}\left(\frac{QK^T}{\sqrt{d}}\right) V \tag{2}$$

where $Q = W_q Z$ is the query features projected from the latent features by the pretrained matrix $W_q$, and $K = W_k c_t$, $V = W_v c_t$ are key and value features projected from the text features $c_t$ with their corresponding pretrained matrices $W_k$, $W_v$.

### 3.2 TASK DEFINITION

We formally define the task of layout-guided multi-subject customization as follows. Given a global text prompt $P$ describing the desired scenario, and a set of subject image–layout pairs $D = \{(I_j, B_j)\}_{j=1}^n$, where $I_j$ is the reference image of subject $S_j$ and $B_j$ is a bounding box specifying its target position, the goal is to synthesize a compositionally coherent image $I^G$ that simultaneously achieves: 1) Textual alignment with the text prompt $P$. 2) Subject identity preservation for each $S_j$. 3) Layout control by placing each subject at its designated location $B_j$.

### 3.3 APPROACH

**Overview.** We now introduce our training-free framework for layout-guided multi-subject customization. The core idea is to decouple the integration of three input conditions — *textual* (text prompt), *visual* (subject images), and *layout* (bounding boxes) — through a bottom-up dual-level attention decoupling strategy. As illustrated in Figure 3, the generation of $I^G$ begins with a randomly initialized latent $z_T \sim \mathcal{N}(0, \mathbf{I})$, which is progressively denoised into $z_0$ by the diffusion process. During denoising, **a)** the *global decoupling* separates cross-attention between text and subject images, and **b)** *local decoupling* further disentangles image cross-attention based on layout constraints, ensuring that each spatial region attends only to its designated subject. Meanwhile, AnyMS employs

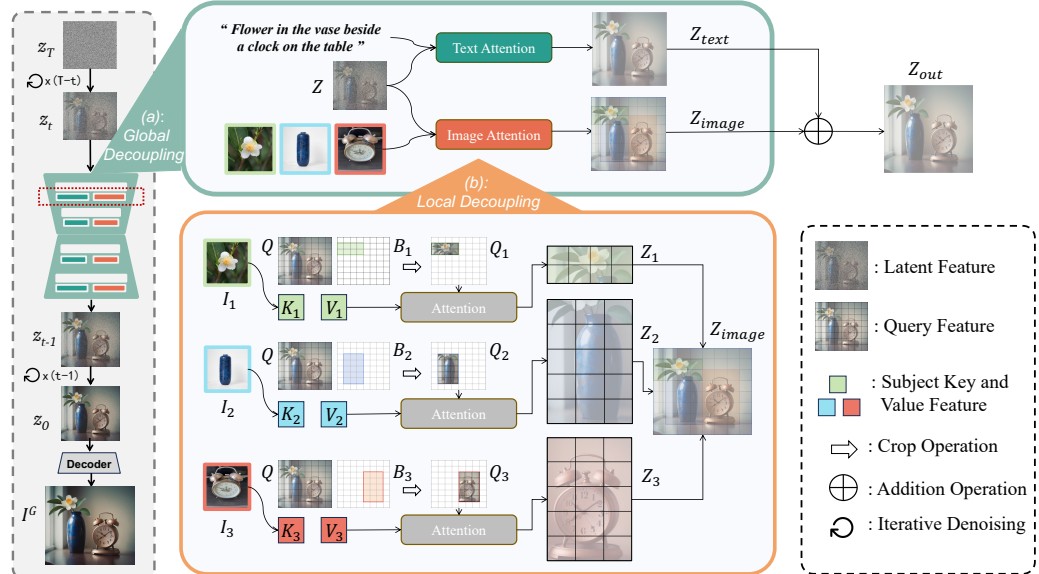

Figure 3: **The Overview of Pipeline.** AnyMS applies a dual-level attention decoupling strategy alongside the general denoising process of the diffusion model. (a) The global decoupling separates cross-attention between text and subject images. (b) The local decoupling further disentangles image cross-attention based on layout constraints. The final $z_0$ is then decoded back to target image $I^G$.

a pre-trained image adapter (Ye et al., 2023; Li et al., 2024) to extract subject-specific features. Finally, the refined latent $z_0$ is decoded into the target image $I^G$.

**Global Decoupling.** A common practice in existing methods (Ruiz et al., 2023; Liu et al., 2023b; Chen et al., 2024; Zhu et al., 2025) is to introduce a special token to represent the target subject and fine-tune the model to bind the token with specific visual features. These approaches typically concatenate such learned tokens with the text prompt and process them jointly through the text cross-attention. However, this design leads to several issues. First, it inevitably causes conflicts between textual and visual conditions: the subject token is entangled with surrounding textual tokens during denoising, resulting in identity distortion, background leakage, and unintended attribute transfer. Second, when spatial relations are described in the prompt (*e.g.*, "standing in front of"), the shared cross-attention may force the subject token to compete with layout conditions, which further undermines precise position control. To mitigate these conflicts and achieve balanced condition integration, AnyMS introduces *global decoupling* as shown in Figure 3(a), where text prompt and subject images are processed by separate cross-attention streams. In this way, textual semantics and visual identity are disentangled at the global level, enabling the model to preserve subjects faithfully while maintaining accurate text alignment. The output of each cross-attention block is reformulated as:

$$Z_{\text{out}} = Z_{\text{text}} + Z_{\text{image}} \tag{3}$$

where $Z_{\text{text}} = \text{CA}_{\text{text}}(Z, P)$ is obtained through the text cross-attention (*c.f.*,Eq 2). And $Z_{\text{image}}$ is obtained through the image cross-attention, which we will specify below.

**Local Decoupling.** In general image cross-attention (Ding et al., 2024), the entire latent features attend to all subject images simultaneously. This design easily leads to subject–subject conflicts, where visual features from different subjects interfere with each other, causing identity confusion or attribute mixing. To address this issue and better exploit layout constraints, as shown in Figure 3(b), AnyMS further introduces *local decoupling*. Specifically, the bounding boxes are used to restrict the interaction between latent regions and their corresponding subject features, ensuring that each spatial area only attends to its designated subject. We formulate the image cross-attention as:

$$Z_{\text{image}} = \text{CA}_{\text{image}}(Z, \{(I_j, B_j)\}_{j=1}^n) \tag{4}$$

Specifically, the local image cross-attention consists of two steps.

*1) Training-free Subject Feature Extraction.* Instead of fine-tuning the diffusion model to learn new subject embeddings, AnyMS leverages pre-trained image adapters to directly extract subject features

in a training-free manner. Given a reference image $I_j$ of subject $S_j$, the adapter encodes its image features $c_j$, which is already aligned with the diffusion model. This image features is then projected into the subject-specific key and value features, $K_j = W'_k c_j$ and $V_j = W'_v c_j$, using the adapter's pretrained projection matrices $W'_k$ and $W'_v$.

*2) Attention Cropping and Merging.* We begin by initializing $Z_{\text{image}}$ with the global query feature $Q \in \mathbb{R}^{H \times W}$ projected from the latent features. To incorporate subject-specific information while preserving layout constraints, we adopt a crop–and–merge strategy. For each subject $S_j$ with bounding box $B_j = [h_s, h_e] \times [w_s, w_e]$, we extract the corresponding subregion $Q_j = Q[h_s : h_e, w_s : w_e]$. Subject features are then injected into this region via cross-attention:

$$Z_j = \text{Softmax}\left(\frac{Q_j K_j^T}{\sqrt{d}}\right) V_j \tag{5}$$

Afterwards, we merge the local outputs back to their original positions:

$$Z_{\text{image}}[h_s : h_e, w_s : w_e] = Z_j \tag{6}$$

For overlapping regions, we resolve subject–subject conflicts by enforcing a semantic priority order (*e.g.*, attribute > object, and foreground > background), ensuring consistent layout and identity preservation. The resulting $Z_{\text{image}}$ thus captures local subject fidelity and structural coherence. By combining it with text cross-attention outputs $Z_{\text{text}}$, we achieve a balanced alignment between global textual semantics and subject-aware layout.

Finally, this dual-level attention decoupling is applied to every cross-attention layer of the U-Net and across all denoising timesteps during inference, yielding stronger text-subject–layout control with only marginal inference overhead.

## 4 EXPERIMENTS

### 4.1 EXPERIMENTAL SETUP

**Dataset.** For a fair and comprehensive evaluation, we follow the widely recognized subject customization benchmarks and conduct experiments on subjects drawn from DreamBooth (Ruiz et al., 2023), Custom-Concept101 (Kumari et al., 2023), and Textual Inversion (Gal et al., 2022). In total, we collect 29 subjects covering diverse categories, including animals, objects, vehicles, and humans, which ensures broad coverage for multi-subject evaluation. To enrich diversity, we form 11 combinations for the quantitative study and present more combinations for visual display .

**Evaluation Metrics.** We comprehensively evaluate the performance of layout-guided multi-subject customization from three perspectives: *1) Layout control.* We assess the consistency between the generated layout and the input bounding boxes by computing mean Intersection-over-Union (mIoU) and AP@50 scores. Both metrics are measured using pre-trained object detection models GroundingDINO (Liu et al., 2024). *2) Identity preservation.* To evaluate how well the generated images retain subject identity, we extract the target subject regions from generated images via GroundingDINO (Liu et al., 2024) and SAM (Kirillov et al., 2023), and compute similarity with corresponding reference images using multiple image-level similarity metrics, including CLIP-I (Radford et al., 2021), DreamSim (Fu et al., 2023), and DINO (Oquab et al.). *3) Text alignment.* We evaluate the semantic consistency between generated images and text prompts using CLIP-T (Radford et al., 2021) similarity in the CLIP embedding space.

**Baselines.** We compare AnyMS with representative state-of-the-art layout-guided multi-subject customization methods, which can be broadly divided into three categories: *1) Latent Injection.* **MuDI**(Jang et al., 2024) augments training data using OWLv2(Minderer et al., 2023) and SAM (Kirillov et al., 2023), and introduces a latent initialization strategy with bounding boxes to provide a better starting point for inference. *2) Attention Rectifying.* **Cones2**(Liu et al., 2023b) learns subject tokens via finetuning and constrains their cross-attention maps within the assigned bounding box regions. *3) Adapter Tuning.* **MS-Diffusion**(Wang et al., 2024a) trains a layout-aware adapter to jointly encode subject, prompt, and layout inputs, leveraging a carefully curated multi-subject dataset with bounding box annotations. In addition, we also report results of the state-of-the-art layout-free multi-subject customization method, **LatexBlend** (Jin et al., 2025), for reference.

| Model | Layout Control | | Identity Preservation | | | Text Alignment |
|---|---|---|---|---|---|---|
| | AP50↑ | mIOU↑ | CLIP-I↑ | DreamSim↑ | DINO↑ | CLIP-T↑ |
| LatexBlend (Jin et al., 2025) | - | - | 65.90 | 43.37 | 40.30 | 32.40 |
| Cones 2 (Liu et al., 2023b) | 26.18 | 38.10 | 67.50 | 46.00 | 41.71 | 33.49 |
| MuDI (Jang et al., 2024) | 19.63 | 36.08 | 73.24 | 58.87 | 55.19 | 32.36 |
| MS-Diffusion (Wang et al., 2024a) | 32.26 | 48.37 | 72.04 | 59.22 | **57.33** | 34.63 |
| AnyMS (Ours) | **35.65** | **49.75** | **74.46** | **59.62** | 54.64 | **35.82** |

Table 1: **Quantitative Results of Layout-guided Multi-subject Customization. Bold** represent the highest metric. Since LatexBlend is implemented without layout control, it is not strictly comparable and we shade its results in gray.

**Implementation Details.** We implement our method with Stable Diffusion XL (SDXL) (Podell et al., 2023) as the base model and employ IP-Adapter (Ye et al., 2023) as the pretrained image adapter for subject feature extraction. For all baseline methods, we follow their default settings. All tested images are generated with a resolution of 1024x1024.

## 4.2 QUANTITATIVE EVALUATION

**Setting.** We construct 11 experimental cases of multi-subject composition, covering subject counts ranging from 2 to 5, each with diverse layout configurations (*i.e.*, bounding boxes). For every case, we randomly generate 100 images and report the averaged evaluation metrics in Table 1. To further analyze scalability, Figure 4 visualizes the performance trends of different methods under varying numbers of subjects. See more details in Appendix.

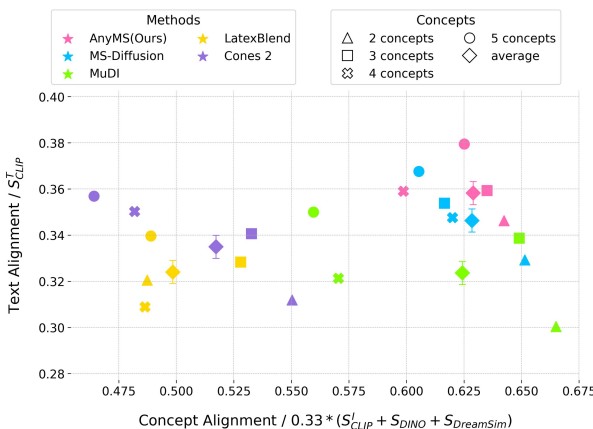

Figure 4: **Detailed Quantitative Results with Different Numbers of Subjects.** Marker shape indicates the number of subjects, and color represents the method used. Rhombus denotes aggregated results.

**Results.** As shown in Table 1, we observe: 1) For layout control and text alignment, AnyMS achieves the highest performance across both metrics, demonstrating accurate spatial control of subjects and faithful adherence to textual descriptions. 2) AnyMS also consistently surpasses baseline methods in identity preservation, achieving superior similarity scores on both CLIP-I and DreamSim, and competitive performance on the DINO score. These results demonstrate that AnyMS achieves a well-balanced trade-off among layout control, text alignment, and identity preservation. 3) Additionally, as shown in Figure 4, AnyMS maintains strong performance when the number of subjects increases from 2 to 5. In particular, our method shows clear advantages under more complex compositions. These results highlight that AnyMS not only supports complex scene composition but also scales effectively to larger numbers of subjects, achieving the best overall performance.

## 4.3 QUALITATIVE EVALUATION

As shown in Figure 5, for multi-subject customization cases with subject numbers varying from 2 to 5, common failure patterns across baselines include object omission, fidelity degradation, and poor prompt alignment. Specifically, MS-Diffusion struggles with complex scenarios and generalizes poorly to unseen subjects, leading to degraded identity reconstruction and inadequate prompt adherence. MuDI prioritizes identity preservation at the expense of prompt compliance and fails to model interactions between objects (*e.g.*, `carrying` and `wearing`). Cones2 can generate multiple subjects but suffers from low identity preservation. In contrast, AnyMS excels in multi-subject customization, producing visually harmonious images with coherent layout by achieving both high

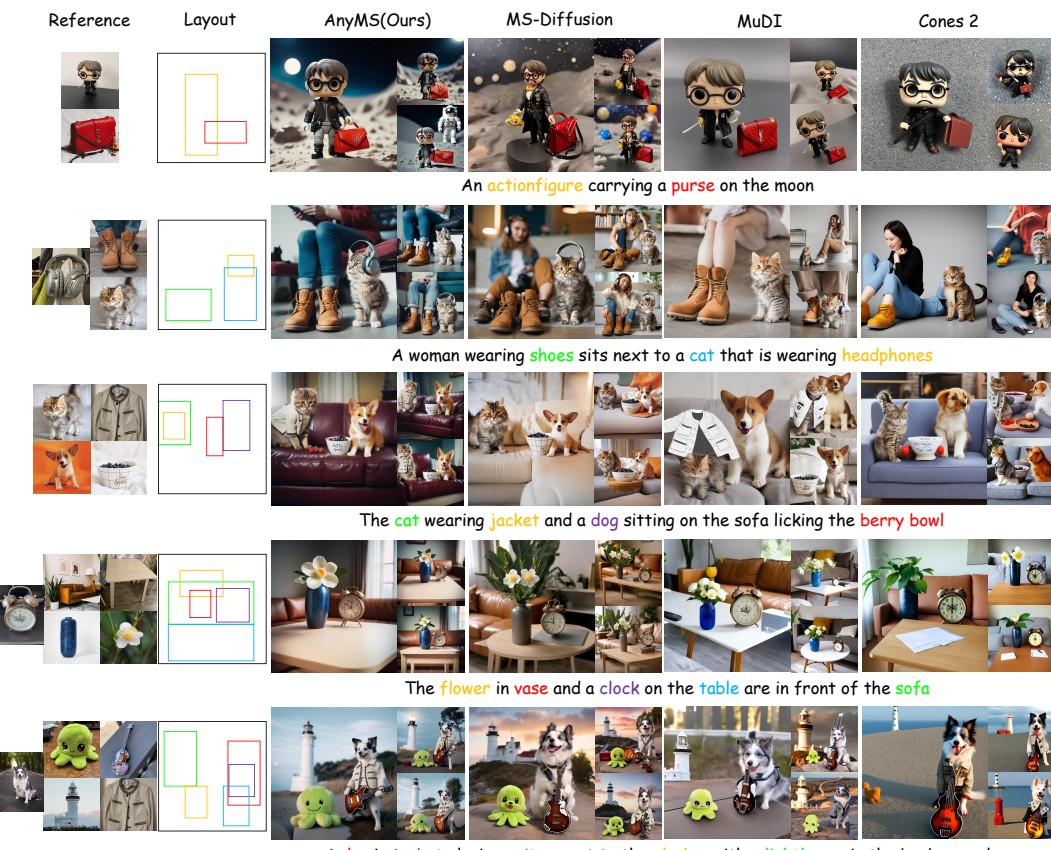

Reference Layout AnyMS(Ours) MS-Diffusion MuDI Cones 2

An actionfigure carrying a purse on the moon

A woman wearing shoes sits next to a cat that is wearing headphones

The cat wearing jacket and a dog sitting on the sofa licking the berry bowl

The flower in vase and a clock on the table are in front of the sofa

A dog in jacket playing guitar next to the plushy, with a lighthouse in the background

Figure 5: **Comparison of Layout-guided Multi-subject Customization.** Different colors show the associations between subjects and their corresponding layout configurations.

fidelity and prompt alignment, while maintaining a balanced trade-off between identity preservation and layout control. Moreover, AnyMS demonstrates robust performance as the number of subjects increases, further highlighting its scalability in handling complex compositions.

## 4.4 ABLATION STUDY

**Settings.** We conduct ablations to validate the effects of the proposed method with two different settings. 1) Remove the crop-and-merge operation in local decoupling. We directly utilize the entire query $Q$ to calculate the image attention $Z_j$ for subject $S_j$, rather than extracting the subregion $Q_j$. We then apply a mask $M_j$ based on the bounding box $B_j$ to $Z_j$ to get layout-aware attention, and add them together to get the final output $Z_{\text{image}}$. The Eq 5 and Eq 6 are reformed as:

$$Z_j = \text{Softmax}\left(\frac{QK_j^T}{\sqrt{d}}\right)V_j, Z_{\text{image}} = \sum_{j=1}^{n}(Z_j \odot M_j) \tag{7}$$

2) Remove the whole local decoupling operation. We further remove all layout guidance by directly adding each $S_j$ to get the final output $Z_{\text{image}} = \sum_{j=1}^{n} Z_j$.

**Results.** As shown in Figure 6, we have the following observations: 1) Removing the crop-and-merge operation works reasonably when the subject number is small (*e.g.*, two subjects), but as the count increases the model struggles to disentangle subject features, leading to issues like subject missing, object fusion (*e.g.*, the castle and the lighthouse), and degraded identity preservation (*e.g.*, the jacket). 2) Without local decoupling, image fidelity drops further, and subject entanglement becomes more severe. In contrast, AnyMS effectively disentangles subject features and preserves high image fidelity. To further validate the effectiveness of AnyMS, we conducted quantitative evaluations on scenarios involving three or more subjects with a total of seven combinations. As

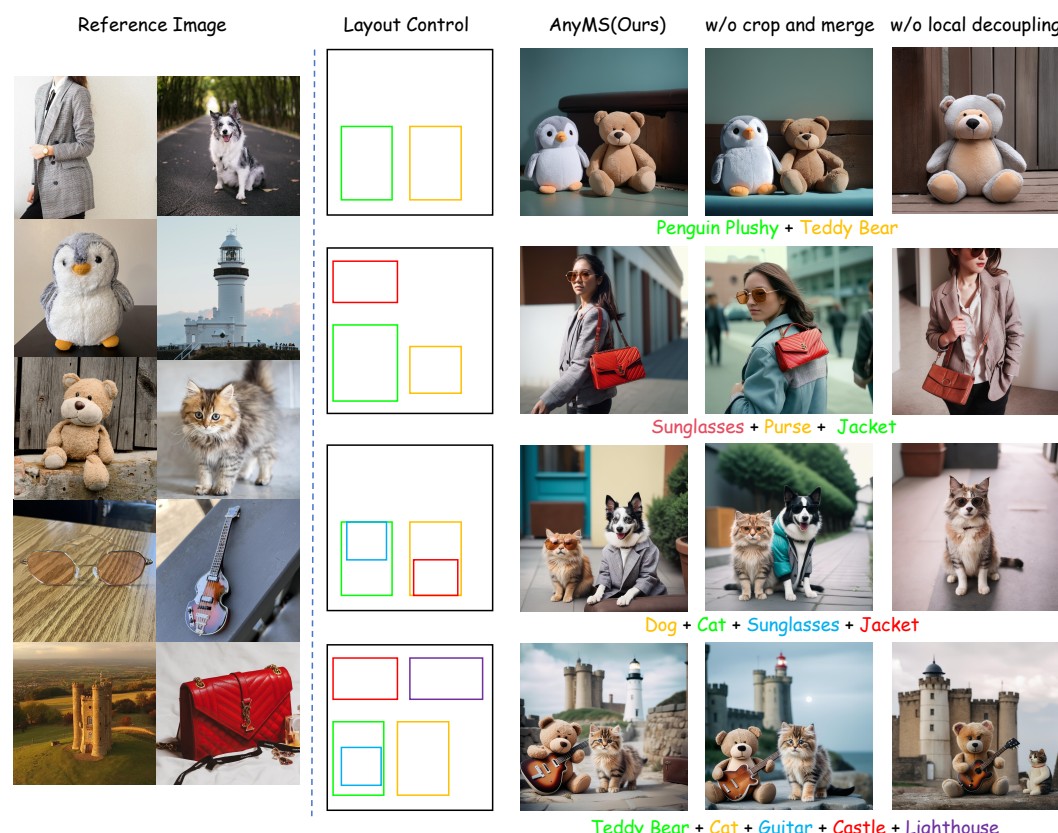

Figure 6: **Ablations Results.** Different colors show the associations between subjects and their corresponding layout configurations. Crop-and-merge plays a crucial role in decoupling features and layout control as the number of subjects increases.

shown in Table 2, AnyMS tops all metrics. With layout guidance, the local decoupling operation further confines subjects to their designated regions, enabling a stable balance between identity preservation and layout control, while scaling robustly to larger numbers of subjects.

| Model | mIOU↑ | CLIP-I↑ | CLIP-T↑ |
|---|---|---|---|
| w/o local decoupling | - | 67.26 | 35.38 |
| w/o crop and merge | 38.78 | 72.13 | 35.67 |
| AnyMS (Ours) | **44.55** | **73.45** | **36.50** |

Table 2: **Quantitative Results for Ablation Study.** Evaluated on seven combinations with more than three subjects.

## 5 CONCLUSION

In this paper, we introduced AnyMS, a novel training-free framework for layout-guided multi-subject customization. By performing bottom-up dual-level attention decoupling, AnyMS effectively disentangles textual, visual, and layout conditions, achieving a better balance among text alignment, subject identity preservation, and layout control. Extensive experiments and ablations validate the effectiveness of AnyMS and its scalability to increasingly complex multi-subject compositions. Moving forward, we plan to 1) extend AnyMS into video customization; 2) explore advanced techniques for customization that jointly consider subject, action, and style.

**Limitations.** While AnyMS achieves strong performance without additional training, its effectiveness still depends on the capacity of the underlying pre-trained diffusion model and image adapters. Consequently, the upper bound on the number of subjects, the complexity of scenes, and the robustness of subject feature extraction may be constrained. Future work could explore integrating stronger foundation models and adaptive feature learning strategies to further enhance scalability and generalization.

ETHICS STATEMENT

This work adheres to the ICLR Code of Ethics. No human subjects or animal experimentation were involved in this study. All reference image datasets we use in our experiments and presentation, including MS-Diffusion, BLIP-Diffusion, Custom-Concept101, DreamBooth, Textual Inversion, are publicly available under relevant research licenses and comply with usage guidelines, ensuring no violation of privacy or ethical standards.

REPRODUCIBILITY STATEMENT

We have made every effort to ensure the reproducibility of our results. The experimental setup, including model architectures, configurations, and reference images are fully detailed in the main text and appendix. Our project can be found in this anonymous link https://anonymous.4open.science/r/AnyMS-659CAJDFQMYBH

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

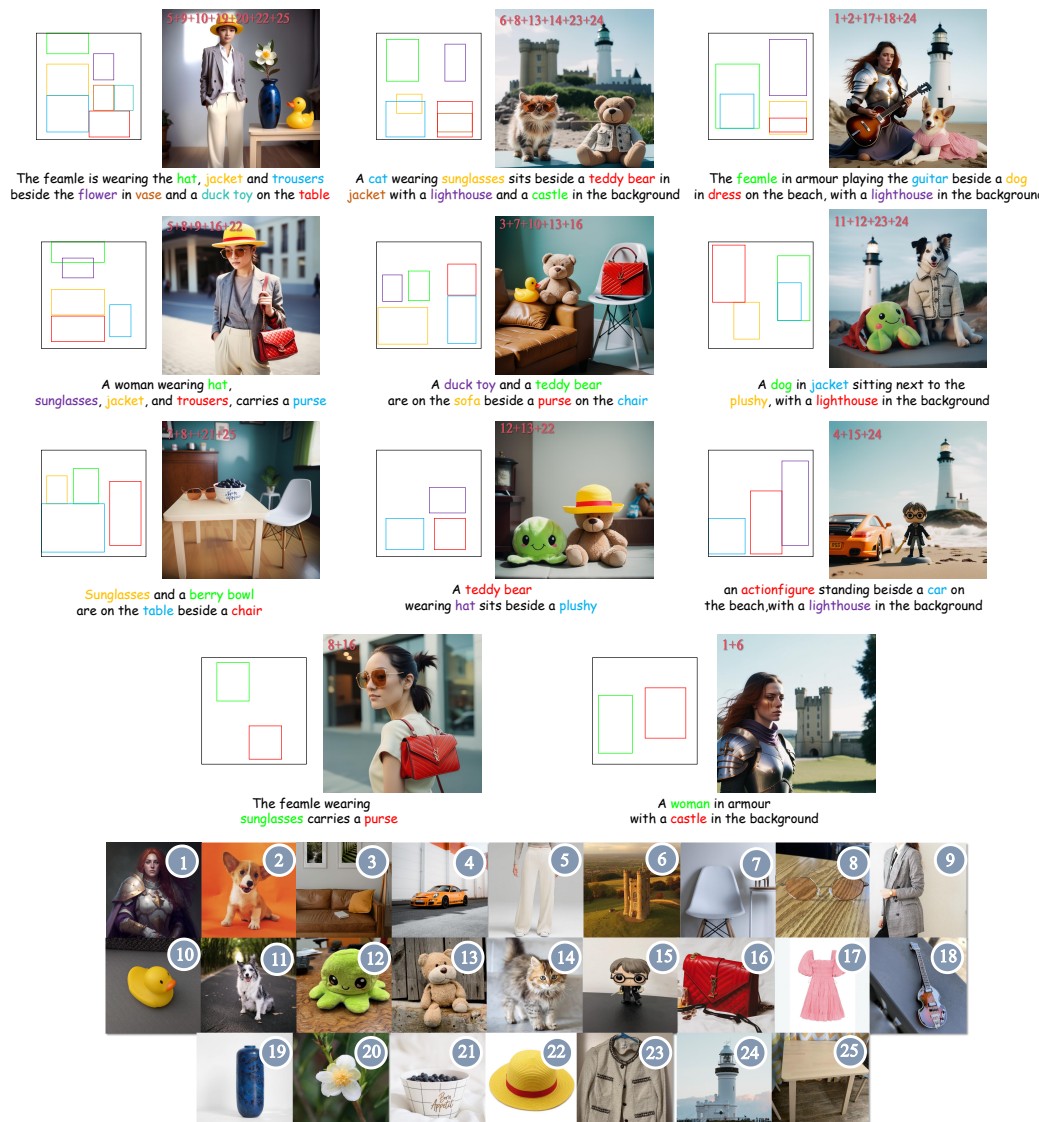

Figure 7: **Visualization Details and Additional Results.** Different colors show the associations between subjects and their corresponding layout configurations. We show more results beyond the first page with these settings.

## APPENDIX

This appendix is organized as follows:

- Section A provides implementation details of Figure 1 and more visualization results.
- Section B provides implementation details of quantitative evaluation.
- Section C provides the broader impact of our method.
- Section D provides the use of LLMs.

## A  VISUALIZATION DETAILS AND MORE RESULTS

We provide implementation details of Figure 1 on the first page, including layout configurations and text prompts, and we also show additional multi-subject customization results based on the settings in Figure 7. The results encompass various combination types and include scenarios with the number

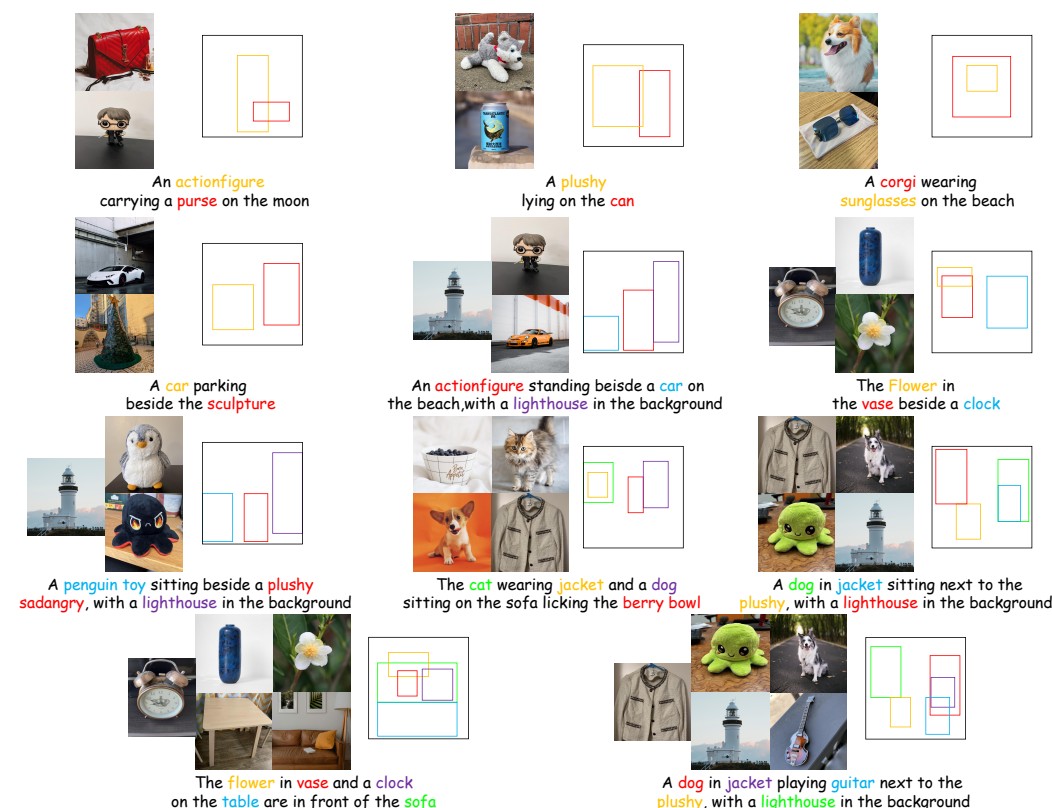

Figure 8: **Details of 11 Combinations in Quantitative Evaluation.** Different colors show the associations between subjects and their corresponding layout configurations.

of subjects ranging from 2 to 7, fully demonstrating the generalizability and robustness of AnyMS. AnyMS has the ability to balance the trade-off among text-alignment, subject identity preservation, and layout control while scaling to a larger number of subjects.

## B  IMPLEMENTATION DETAILS OF QUANTITATIVE EVALUATION

As shown in Figure 8, for the quantitative evaluation, we collect 23 different subjects covering a variety of categories, and form 11 combinations. For each combination, we specify a tuple containing reference images, text-prompt, and bounding boxes. To conduct a comprehensive evaluation, the number of subjects ranged from 2 to 5, assessing the ability of models to decouple subject features as the number of subjects increases. In addition, the ways subjects interact (e.g., carrying and wearing) and the backgrounds (e.g., moon and beach) are also diverse.

## C  BROADER IMPACT

As a multi-subject customization method, AnyMS can generate a text-aligned and well-composed image involving multiple user-provided subjects without training. This means that AnyMS has the potential to play a crucial role in advertising and the film industries: we can create digital doubles of actors and seamlessly render them with virtual cartoon characters into posters for propagation. In addition, generated images with a variety of backgrounds and styles can further enhance diversity, offering audiences a novel experience. AnyMS can also function as entertainment for users in social media. However, there are several points to note when implementing AnyMS. 1) Sensitive terms such as sexual content and political statements should be detected and blocked. 2) Privacy and portrait rights of individuals should be protected. The generated images should be authorized by the person concerned. Overall, AnyMS can enrich entertainment activities for the public with proper application.

## D    LLM USAGE STATEMENT

In this paper, for readability, we strictly utilize LLMs to identify and correct grammatical errors under the supervision of the authors, ensuring clarity and precision of expression throughout the paper.

