# OpenReview forum: "AnyMS: Bottom-up Attention Decoupling for Layout-guided and Training-free Multi-subject Customization"
_ICLR.cc/2026/Conference — ICLR 2026 Conference Withdrawn Submission_

### Official Review · Reviewer_8mr6 · 2025-10-28

**Soundness:** 3
**Presentation:** 3
**Contribution:** 2
**Rating:** 4
**Confidence:** 4

**Summary:**

AnyMS targets layout-guided multi-subject customization: given a prompt, one reference image per subject, and bounding-box layouts, the goal is to synthesize an image that preserves layout and subject identity without additional training. The method performs dual-level attention decoupling during diffusion inference. At the global level, it separates cross-attention into a text stream and an image (reference) stream to reduce competition between description tokens and subject features, improving text alignment. At the local level, it enforces per-box attention via a crop-and-merge procedure, so each spatial region attends only to its designated subject, mitigating identity leakage and boundary spillover. Built on SDXL with a pre-trained image adapter (e.g., IP-Adapter), the pipeline is training-free and operates purely at generation time.

**Strengths:**

Overall, the paper has a clear and practical focus on layout guided multi subject customization. The design goes straight at the usual failure modes: identity leakage, boundary spillover, and layout drift. The dual level attention decoupling runs only at inference and, paired with a pretrained image adapter, gives a training free pipeline that keeps untouched regions stable while enforcing per box control. Empirically, it improves layout control, identity preservation, and text alignment on compositions with two to five subjects, with more graceful degradation as the count grows. The ablations are tidy and attribute gains to both global and local decoupling and the crop and merge step. Implementation details are clear enough to reproduce, and the recipe is simple, portable, and low overhead.

**Weaknesses:**

1. There is little analysis of noisy or overlapping boxes, adapter or depth feature noise, conflict resolution when boxes intersect, scaling beyond five subjects, and the runtime or memory cost at common resolutions.
2. The method reads as an assembly of known pieces such as cross attention control, per box masking, and image adapters. The paper does not clearly isolate a new technical insight beyond this integration.
3. Results rely solely on detector/CLIP-based automatic metrics; without a user study, it’s unclear whether the reported gains match human-perceived layout faithfulness, identity fidelity, or prompt adherence.
4. The paper claims “only marginal inference overhead” but reports no latency/VRAM/FLOPs numbers or runtime comparisons with baselines.

**Questions:**

See Weaknesses.

---

### Official Review · Reviewer_PkZj · 2025-11-01

**Soundness:** 3
**Presentation:** 3
**Contribution:** 3
**Rating:** 6
**Confidence:** 5

**Summary:**

This paper introduces AnyMS, a novel training-free framework for layout-guided multi-subject image customization. The authors identify a core trade-off in existing methods between three competing goals: text alignment, subject identity preservation, and layout control. They also note the inefficiency of methods requiring additional training. Experiments reportedly demonstrate that AnyMS achieves state-of-the-art performance, excelling in complex compositions and scaling effectively to a larger number of subjects.

**Strengths:**

1. The results for flexibly composing a varying number of subjects are very impressive.
2. The core innovations and contributions are presented clearly and effectively.

**Weaknesses:**

1. The core contribution of the paper is the "local decoupling" attention mechanism. However, the experimental section lacks a detailed analysis of it. Specifically, the paper would be strengthened by a comparison with other methods that also focus on improving attention mechanisms for customized generation, as well as visualizations of the proposed attention maps to demonstrate its effectiveness.
2. There appears to be a significant overlap in the information presented in Figure 4 and Table 1, as both seem to display the performance metrics of different methods across various objects. The authors should clarify the unique contribution of each or consider consolidating them to improve readability.

**Questions:**

1. The presentation of Figure 3 is confusing. Specifically, the location of 'Local Decoupling' in the network diagram is not clearly marked. This needs to be explicitly pointed out to enhance readability.

---

### Official Review · Reviewer_1rQB · 2025-11-01

**Soundness:** 2
**Presentation:** 3
**Contribution:** 2
**Rating:** 4
**Confidence:** 4

**Summary:**

The authors propose a training-free approach for object customization by disentangling attention maps in diffusion models, which is an interesting and practical direction. They leverage subject-specific features from adapters of existing methods to assist in identity preservation. Experimental results demonstrate that the proposed method achieves a trade-off across various evaluation metrics.

**Strengths:**

- This problem of generation of layout-guided and training-free multi-subject customization is an impoartant task.
- The writing is overall clear and easy to follow.

**Weaknesses:**

- Incremental contribution: The work lies within a line of research that includes numerous recent studies on text alignment, ID preservation, and layout control. This work is particularly in the context of training-free methods based on attention manipulation. Thus, the novelty appears incremental given the existing works.
- Questionable trade-off between tasks: The assumption that text alignment, subject identity preservation, and layout control inherently require a performance trade-off is debatable. Prior works (e.g., MS-Diffusion and MuDI) have demonstrated that these objectives can be mutually beneficial and simultaneously improved. The observed trade-off in this paper likely stems from suboptimal fusion of different attention maps. The authors should include ablation studies to analyze the contribution of each attention component and justify their fusion strategy.
- Missing failure cases: The paper would benefit from a discussion of failure cases to better illustrate the limitations of the approach and guide future research.
- Insufficient layout visualization in the main figure: The first figure (likely the overview or key result) does not clearly show the method’s capability in layout control, which is one of the claimed contributions. Including a more illustrative example would strengthen the presentation.
- The code provided is empty, only two images available.

**Questions:**

See weakness.

---

### Official Review · Reviewer_rt2m · 2025-11-02

**Soundness:** 2
**Presentation:** 2
**Contribution:** 2
**Rating:** 4
**Confidence:** 4

**Summary:**

This paper proposes a multi-subject customization method that designs global and local decoupling mechanisms to ensure both text alignment and spatial control.

**Strengths:**

1. The paper is clearly written and easy to follow.
2. The figures and tables are clear and easy to understand.
3. The authors use many formulas to help readers better understand the concepts.

**Weaknesses:**

1. The motivation for the global decoupling design is insufficient. In lines 246-251, the authors claim that concatenating the subject token with the text prompt leads to various issues. However, this claim lacks persuasiveness as the root causes of these issues are not adequately analyzed (e.g. attention map). This further undermines the motivation for the global decoupling design. If I understand correctly, this component is essentially an IP-Adapter, which reinforces the concern about the lack of novelty.
2. Regarding the local decoupling: This part is the key design of the paper; however, I have the following questions:
    (a) Since this is a work on customization synthesis, preserving the identity (ID) of the reference image is crucial. The authors propose using an off-the-shelf image encoder to encode the reference image. Is the extracted image feature more biased towards high-level semantic information or low-level visual details?
    (b) The authors crop a continuous latent feature map. Can the cropped features maintain their original structure in the same way as cropping in pixel space?
If these issues are not well-considered, the entire design of this section becomes questionable.
3. The paper does not compare its method with GroundingBooth.[1]
    [1] Xiong et al. GroundingBooth: Grounding Text-to-Image Customization. Arxiv 2409.08520

**Questions:**

1. Is the method proposed in this work training-free? If so, for Equation (3), does simply adding the two latents together actually work? Is linear interpolation not required?
2.  After encoding an image (3, H, W) into a latent code (c, h, w), if we crop a patch (c, h1, w1) from the latent space and then decode it with a VAE, what will the result be? Will it be identical to the corresponding region in the original image?

---

### Note · Authors · 2025-11-13

I have read and agree with the venue's withdrawal policy on behalf of myself and my co-authors.